# Application of Deep Learning in Histopathology Images of Breast Cancer: A Review

**DOI:** 10.3390/mi13122197

**Published:** 2022-12-11

**Authors:** Yue Zhao, Jie Zhang, Dayu Hu, Hui Qu, Ye Tian, Xiaoyu Cui

**Affiliations:** 1College of Medicine and Biological Information Engineering, Northeastern University, Shenyang 110169, China; 2Key Laboratory of Intelligent Computing in Medical Image, Ministry of Education, Shenyang 110169, China; 3Key Laboratory of Data Analytics and Optimization for Smart Industry, Northeastern University, Shenyang 110169, China

**Keywords:** deep learning, breast cancer, pathological image, histopathology

## Abstract

With the development of artificial intelligence technology and computer hardware functions, deep learning algorithms have become a powerful auxiliary tool for medical image analysis. This study was an attempt to use statistical methods to analyze studies related to the detection, segmentation, and classification of breast cancer in pathological images. After an analysis of 107 articles on the application of deep learning to pathological images of breast cancer, this study is divided into three directions based on the types of results they report: detection, segmentation, and classification. We introduced and analyzed models that performed well in these three directions and summarized the related work from recent years. Based on the results obtained, the significant ability of deep learning in the application of breast cancer pathological images can be recognized. Furthermore, in the classification and detection of pathological images of breast cancer, the accuracy of deep learning algorithms has surpassed that of pathologists in certain circumstances. Our study provides a comprehensive review of the development of breast cancer pathological imaging-related research and provides reliable recommendations for the structure of deep learning network models in different application scenarios.

## 1. Introduction

Cancer is a huge public health problem worldwide. Among cancer types, breast cancer (BC) is the most common cancer in women [1]. Since the late 1970s, the number of breast cancer patients worldwide has increased, and to date, breast cancer has become one of the types of cancer with the highest incidence and mortality rates in the world. Based on statistics from the World Health Organization, 8.8 million people died of cancer in 2020, of which 684,996 died of breast cancer [2]. A histopathological examination of breast cancer is the “gold standard” for a breast cancer diagnosis [3]. Pathologists can distinguish between normal tissue, non-malignant (benign) tissue, and malignant lesions by observing the microscopic structure and organization of biopsy samples in microhistological images.

A traditional pathological diagnosis has high prestige in a medical diagnosis. The pathologist observes tissue slices through a microscope and makes the corresponding cancer diagnosis by observing the tissue structure and cytopathic characteristics of the slices. The staining density and flatness of the slice, as well as the collection and storage of pathological slice images, may affect the integrity of the final pathological slice image. The inherent complexity and diversity of breast histological images make the diagnostic work of pathologists tedious and time-consuming. Additionally, differences in experience and the subjectivity of pathological diagnostic criteria often lead to inconsistency and the non-reproducibility of diagnostic results [4]. The development of digital pathology has reduced these effects, which is helpful in obtaining high-resolution images [5]. Compared to traditional pathology, digital pathology uses digital pathology systems to digitize and network process pathological resources. With the application of big data technology in the medical field, we can describe the collection visualization, long-term storage, and synchronous browsing of data. The processing of collected pathological resources is no longer restricted by time and space. Therefore, digital pathology has become widely used in related fields of pathology [6].

In recent years, the continuous development of artificial intelligence technology has achieved remarkable results in various fields. With the continuous improvement and increase in medical equipment and data recording systems, the medical field has several large-scale datasets, such as Camelyon16. As a subcategory of artificial intelligence, deep learning has also benefited greatly and achieved remarkable achievements [7]. Srinidhi et al., presented a comprehensive review of state-of-the-art deep learning approaches that have been used in the context of histopathological image analysis [8]. Wang et al., summarized recent deep learning approaches relevant to precision oncology and reviewed more than 150 articles from the past six years [9]. We searched original articles published from 2007 to 2022 using “breast cancer”, “pathology”, and “deep learning” as keywords in Web of Science and performed a statistical analysis. By analyzing the studies from 2007 to 2022, we found that the articles about deep learning have gradually increased since 2007, Figure 1A. Robertson et al., introduced the development from image processing technology to artificial intelligence in breast pathology [10]. Gao et al., introduced medical image analysis technology based on convolutional neural networks in computer-aided diagnosis (CAD) research [11]. Biswas et al., introduced the development of deep learning and certain applications in medical imaging [12]. Figure 1B shows the summary of the themes of review-type articles in articles on the application of deep learning to breast cancer pathological pictures. Although some articles summarizing the application of breast cancer pathology in deep learning have been published [13,14,15], with the rapid development of deep learning algorithms, it is still necessary to systematically and comprehensively summarize the research results in this field. In this review, we summarize 202 articles from Web of Science from 2007 to the present on the application of deep learning in breast cancer pathology. This review analyzed the application of deep learning methods in the detection, segmentation, and classification of breast pathological images. In addition, relevant articles were introduced and sorted, respectively, and common public breast cancer pathological image datasets are summarized.

Section 2 of this paper comprehensively summarizes the public dataset related to pathological images of breast cancer. Section 3 explores the application of the deep learning method in breast cancer pathological images from the three perspectives of detection, segmentation, and classification. Section 4 summarizes the above methods and looks forward to the future development of this field.

## 2. Datasets

Most deep learning networks used to process pathological images are algorithms with supervision. To improve the network accuracy, a large amount of labeled data is required for training to improve the fit. The acquisition and labeling of datasets are time-consuming and complex [16]. In this section, we summarize the data and information contained in the existing breast cancer public datasets and the number and types of pathological images they contain. Table 1 summarizes the commonly used public datasets. Breast cancer histopathological image classification (BreakHis) consists of 9109 microscopic images of breast tumor tissue from 82 patients using different magnification scales (40×, 100×, 200×, and 400×). The Cancer Imaging Archive (TCIA) has identified and hosted a vast archive of medical images of cancer for public download and includes typical patient images related to common diseases, imaging modalities or types (Magnetic Resonance Imaging (MRI), Computed Tomography (CT), digital histopathology, etc.), or research priorities. The primary file format for TCIA is Digital Imaging and Communications in Medicine (DICOM). The Genomic Data Sharing Area (GDC) is a research initiative of the National Cancer Institute (NCI). The mission of the GDC is to provide the cancer research community with a unified data repository to share data across cancer genome research to support precision medicine. Sklearn datasets are composed of classic data that can be directly called by the classic machine learning module Sklearn in Python; these datasets include Boston house price data, Wisconsin breast cancer data, diabetes data, the handwritten digital dataset, Fisher’s iris data, and wine data. BACH (International Conference on Image Analysis and Recognition (ICIAR) 2018 Grand Challenge) refers to the ICIAR Grand Challenge on BreAst Cancer Histology images. The challenge provides a dataset consisting of the histological microscopic examination of the breast stained with hematoxylin and eosin (H&E) and the entire slide image. Camelyon16 refers to the dataset provided by the Camelyon16 Challenge. The goal of this challenge is to evaluate new and existing algorithms to automatically detect metastases in whole-slide images of H&E-stained lymph node sections. The data in the challenge included a total of 400 full-slide images (WSIs) from sentinel nodes from two separate datasets from the Radboud University Medical Centre (Nijmegen, Netherlands) and the University Medical Centre of Utrecht, Netherlands. CAMELYON17 is a dataset provided by the CAMELYON17 Challenge. The goal of this challenge is to evaluate new and existing algorithms for the automatic detection and classification of breast cancer metastases in full-slide images of histological lymph nodes. The CAMELYON17 dataset comes from five medical centers in the Netherlands. The WSIs are available as TIFF images. Comments on the lesion level are provided in the form of XML files. For training, 100 patients will be selected, and another 100 will be tested.

## 3. Methodology

In 2006, the concept of deep learning again attracted the attention of researchers [17]. From 2007, deep learning began to be applied in breast pathology image data. Then, under the efforts of many researchers, random gradient descent (SGD), Dropout, and other network optimization strategies were successively proposed, especially GPU parallel computing technology that solves the problem of multiple optimization times of deep network parameters for a long time, which has set off an upsurge of deep learning worldwide and continues to this day. Over the past 10 years, many classical deep learning architectures were proposed, such as AlexNet [18], Recurrent Neural Networks (RNN), Long Short-Term Memory (LSTM) [19], Generative Adversarial Network (GAN) [20], and transformer [21]. In this section, we investigate the application of deep learning to histopathological sections of breast cancer, including the most advanced and effective models available today, and we provide a summary of related work. The literature can be divided into three primary categories based on the direction of research reported in each article: breast cancer detection, breast cancer segmentation, and breast cancer classification [22].

Figure 2 shows the common basic neural network structure, where yellow squares represent convolution layers, orange squares represent pooling layers, purple squares represent full-connected layers, and blue squares represent deconvolution layers. Among them, Figure 2A is a simplified convolution neural network, which is composed of only two layers of convolution, and the rest of the depth convolution neural network can be composed by superimposing Figure 2A. LeNet (Figure 2B) [23] is an early convolutional neural network, which was proposed by Yann LeCun et al., in 1990. AlexNet (Figure 2C) is a deep convolutional neural network proposed by Hinton et al., in 2012 and won the championship in the ImageNet challenge that year. Figure 2D is a full convolution neural network (FCN) used for semantic segmentation in the early stage [24]. It is also widely used in the early research of breast cancer pathological image segmentation task. The appearance of U-Net [25] has significantly improved the performance of FCN in medical image segmentation tasks. Holistically Nested Network (HED) [26] has achieved better results than traditional edge detection algorithms in edge detection tasks, and this method has also been used to improve the performance of breast cancer segmentation tasks [27].

### 3.1. Detection of Breast Lesions

In medical image analysis, detection aims to locate areas of interest in tissue slices [28,29,30,31,32,33,34,35,36,37,38,39,40]. The detection system provides strong support for object segmentation, the distinction between malignant and benign tumors, or the detection of tumors or lesions. For example, nuclear or mitosis has important implications for cancer screening. Cell spatial distribution analysis and mitotic count provide support for differentiation. Automatic cell/nucleus detection is a prerequisite for a series of subsequent tasks, such as cell/nucleus instance segmentation, tracking, and morphological measurements [41]. In recent years, many studies based on deep learning were performed in this field of study. Among existing deep learning detection algorithms, CNN-based networks perform better than other network structures in detection accuracy [42]. In certain areas, CNN-based networks have achieved diagnostic standards that surpass pathologists [43]. Next, we will introduce some typical models that work particularly well for accuracy and performance. Table 2 shows the application of deep learning algorithm in histopathological detection of breast cancer. The application of deep learning algorithm in histopathological detection of breast cancer was classified according to the types of models, and the strategies were summarized.

George et al. [44] proposed a low-complexity breast cancer detection convolutional neural network (CNN) called NucDeep, which includes a low-complexity CNN for feature extraction of non-overlapping nuclear plaques and converts local nuclear features into compact image-level features to improve classifier performance (Figure 3A). Chen et al. [45] proposed a novel deep cascaded neural network model (CasNN). CasNN greatly increased the speed of detection of mitosis (Figure 3B). Liu et al. [43] proposed InceptionV3 as a way to automatically detect and locate cancers in high-resolution images (Figure 3C). The author uses InceptionV3 as an experimental framework, and the balance and expansion of data are achieved through data enhancement and balance to improve model accuracy. Bardou et al. [46] proposed a method for automatic classification of breast cancer histological images based on a convolutional neural network (Figure 3D). The linear rectification function (ReLU) layer is used for the convolution and fully connected layers to accelerate the convergence learning rate, introduce nonlinearity, and adjust the network weight to prevent overfitting.

Some classical detection algorithms [47,48,49] in the field of natural images have also been used in the problems related to pathological images of breast cancer. Lu et al. [50] proposed a model based on the latest yolo v4 structure that can quickly and accurately segment the lesion area in high-resolution breast cancer pathological slices. The ROI recognition accuracy is 0.936 and F1 score is 0.787, which is of great significance for improving the diagnostic efficiency and accuracy of pathologists on breast pathological images. Huang et al. [51] proposed an algorithm for breast cancer pathological image nuclear detection based on mask RCNN. This method effectively combines feature pyramid network (FPN), ResNet, and other modules to achieve more accurate detection. Harrison et al. [52] proposed an algorithm for tumor detection in breast pathological images based on Faster RCNN and found that patching the images can significantly improve the sensitivity of the model, from 1% to 60%, and the performance improvement brought by dye normalization is limited. Yamaguchi et al. [53] proposed an automatic detection algorithm for breast cancer based on single-shot multibox detector (SSD) and achieved 88.3% and 90.5% diagnostic accuracy in two (benign or malignant) and three (benign, non-invasive carcinoma, or invasive carcinoma) classification tasks, respectively. Mitotic cell count is an important biomarker for grading and prognosis of breast cancer and is also a common application of pathological intelligence analysis of breast cancer. Zorgani et al. [54] designed a method to detect breast cancer mitotic cells based on the deep yolo architecture and obtained 0.839 F1 measure on the ICPR2012 dataset.

Thus, by reading and analyzing the article, we found that the model proposed by [44] is a low-complexity model with classification results comparable to existing technologies and uses nucleus patches alone rather than random patches. The proposed method of [45] can obtain various multi-level and multi-scale features from breast cancer histopathological images, providing competitive performance in the classification of complex breast cancer histopathological images. However, the dataset collected is relatively small and contains only two types of images, and the dataset should be extended to include images for multiclass classification problems. The authors of [43] made data enhancement for negative samples to solve the large gap between positive and negative samples and optimized the sampling process to remove the patch as the background. After the probability graph is obtained, the tumor region is continuously iterated according to the current maximum value to predict the tumor region based on the non-maximum suppression method. In [46], compared with CNN, the performance of the manual feature-based method based on encoding model for local descriptors to construct image representation is low, and the performance of multiclass classification is lower than that of binary classification. Classical object detection algorithms can also achieve remarkable results in the field of pathological images, especially Faster RCNN series algorithms.

From the above results, it can be seen that the deep learning model has the advantages of direct learning characteristics on breast cancer pathological images, which can greatly reduce the manual investment and also reduce the artificial differences caused by manual reading. Higher accuracy also provides help for the development of precision medicine.

**Table 2 micromachines-13-02197-t002:** Summary of the application of deep learning algorithms in breast cancer histopathology for detection.

Model	Strategy	Advantages	Publication
RNN	Development of decision support systems for pathology	RNN allows neurons in the hidden layer to communicate with each other, storing the previous output as information in the hidden layer	[55]
	Propose a SmallMitosis framework for the detection of mitotic cells from hematoxylin and eosin (H&E)-stained breast histological images		[56]
Inception	Histologic identification of tumor cells in lymph nodes	Inception increases the width of the network by pooling each layer with a different convolution to extract features from the previous layer, and by adding a 1*1 convolution after the pooling layer before the 3*3 and 5*5 convolutions, which effectively avoids complex parameters and computational effort	[57]
	Improve the computer-aided diagnosis method based on deep learning		[58]
ResNet	Detection of invasive ductal carcinoma in breast histological images and the classification of lymphoma subtypes	The main feature of ResNet is the residual block, the purpose of the residual block is to preserve the characteristics of the parameters before the current layer is trained and to pass these parameters into the subsequent layers together with the trained data	[59]
	Diagnostic breast cancer whole-slide tissue images		[60]
	Propose an automatic detection method for invasive ductal carcinoma (IDC) based on deep transfer learning technology		[61]
	Propose Mask RCNN, a multi-task deep learning framework for object detection and instance segmentation, to automatically detect mitosis		[62]
DCNN	Propose an accurate method for detecting the mitotic cells from histopathological slides using a multi-stage deep learning framework		[63]
	Present an SSAE for efficient nuclei detection on high-resolution histopathological images of breast cancer		[64]
	Introduce deep learning as a technique to improve the objectivity and efficiency of histopathologic slide analysis		[65,66,67,68,69,70]
Semi-Supervised Learning	Present a semi-supervised deep learning strategy for breast cancer diagnosis	Semi-supervised learning is to use a large number of unlabeled samples and a small number of labeled samples to train the classifier, solving the problem of insufficient labeled samples	[71,72]
YOLO	A fast lesion detection method based on yolo is proposed	Simple structure and fast speed	[50]
Faster RCNN	A fast detection method of breast tumor based on Faster RCNN is proposed	Faster RCNN realizes object detection performance with high accuracy through second-order network and Region Proposal Network	[52]
Single Shot multibox Detector (SSD)	An automatic detection method of breast cancer lesion based on SSD is proposed	One stage, good at detecting small objects	[53]

### 3.2. Segmentation Method of Breast Pathological Image

Segmentation refers to dividing the input image into many specific areas with unique properties and extracting them, separating the content in a region of interest (ROI) from the image background. The ROI in a pathological image of breast cancer is part of a lesion. When using deep learning correlation methods, it is generally necessary to analyze and extract the characteristics of tumor lesions in ROI so as to detect and classify pathological images. Pathological image segmentation plays an important role in the field of pathological image processing and analysis, which is helpful to provide reliable basis for clinical auxiliary diagnosis and treatment. Despite the high complexity of pathological images and the lack of simple linear features, pathological image segmentation technology has made significant progress due to the effective application of deep learning algorithm in pathological image segmentation. In pathological image segmentation, deep learning algorithms have made remarkable achievements. Most pathological image segmentation uses supervised deep learning algorithms, such as FCN, RNN, and U-Net. Next, we will introduce some typical models that work particularly well for accuracy and performance. Table 3 shows the application of deep learning algorithm in histopathological segmentation of breast cancer. The application of deep learning algorithm in histopathological segmentation of breast cancer was classified according to the types of models, and the strategies were summarized.

Mehta et al. [73] introduced a method to generate distinguishable tissue-level segmentation masks for breast cancer diagnosis (Figure 4A). This Y-Net network expands and generalizes U-Net, adds a parallel branch for the generation of discriminative maps, and supports modularization of convolution blocks. Guo et al. [74] proposed v3-DCNN, a fast cancer region segmentation framework (Figure 4B). The classification model Inception-V3 was used to preselect the tumor area, and then the semantic segmentation model DCNN was used to segment the 1280 × 1280 patch to reduce computation time and improve accuracy. Pan et al. [75] proposed an automatic nuclear segmentation method for histopathological images of breast cancer stained with H&E. The sparse reconstruction method is used to roughly remove the background to emphasize the core of the pathological image, and then the deep convolutional network (DCN) of the multilayer convolutional network cascade is used to effectively segment the core from the background (Figure 4C). Maria Priego-Torres et al. [76] presented a processing pipeline for automatic segmentation of breast cancer images to present different types of histopathological patterns (Figure 4D). The deep convolutional neural network (DCNN) and the encoder–decoder with separable convolution structure were used to complete the segmentation of each patch, and the local segmentation results were merged based on the effective full connection condition random field (CRF) to avoid discontinuity and inconsistency.

Transformer-based methods are also widely used in medical image segmentation [21]. However, there are few studies about the segmentation task of breast pathological images. Therefore, we try to retrieve transformer-based segmentation methods in the fields related to breast pathological image segmentation, aiming to promote the development of transformer-based methods in this field. Cam et al. [77] quantitatively evaluated the segmentation performance of six popular transformer-based segmentation networks on pathological images based on the PAIP liver histopathology dataset and compared the classical CNN-based segmentation networks. The results show that the transformer-based segmentation network is generally better than the CNN-based model, proving the effectiveness of the transformer architecture on pathological image segmentation tasks. Li et al. [78] proposed a vision language medical image segmentation model, LViT (Language measures Vision Transformer), to solve the problem of insufficient annotation of medical images and verified the cell segmentation performance of this method on the MoNuSeg dataset. Diao et al. [79] introduced transformer into the classic U-Net architecture to extract and encode global context information and achieved SOTA performance in the nasopharyngeal carcinoma pathological image dataset.

Semi-automatic segmentation algorithm also attracted much attention in the field of breast cancer image analysis, mainly applied to X-ray [80], ultrasound [81], MRI [82], and other images, and there is less research on pathological images of breast cancer. In recent related research, Lai et al. [83], in conjunction with semi-supervised and active learning, proposed a segmentation algorithm for brain tissue pathological images and achieved IoU scores competitive with fully supervised learning.

Thus, through research and analysis, we found that the features generated by the discriminant segmentation mask used by the authors in [73] were able to achieve the same segmentation accuracy as the most advanced methods while learning fewer parameters. However, this paper only studied breast biopsy images and did not extend to other medical imaging tasks. The method proposed in [74], based on the V3 DCNN model, achieved a higher FROC score of 83.5% than the champion method Camelyon16 80.7%, and further, the automatic heat map generation of WSI was achieved. However, the proposed model lacks dataset validation and should be tested on more breast histopathological images. In [75], k-SVD and Batch-OMP algorithms were used for sparse reconstruction to enhance the nuclear region. In the segmentation stage, DCN trained by structural label was used to obtain the exact pixel of the nucleus, and morphological operation and some prior knowledge were introduced to improve the segmentation performance and reduce errors. The proposed algorithm is a general method and can be applied to many pathological applications. However, the number of datasets is too small, and the number of background pixels is far more than that of nuclei, so there is an imbalance between the number of nuclear pixels and background pixels. The proposed segmentation model in [76] performed well on standard success rate and similarity segmentation metrics, especially considering that the dataset included WSI images with high tumor variability. Web-based viewers and annotation tools were developed to allow collaboration with pathologists and technologists to establish a way to create datasets. However, all images used in this working study were stained in the same laboratory and digitized using the same scanner, and images from other sources should be used to increase the heterogeneity of the training set. Transformer-based methods have achieved remarkable results in the field of computer vision and become popular in medical image analysis tasks. [21] Transformer can well encode context characteristics. This is also used by most researchers to build the global features of breast cancer pathological images so as to improve the model performance. A large number of experimental results show the effectiveness of transformer architecture in this field [77,79,80]. The semi-automatic segmentation algorithm is relatively less used in the breast cancer pathological image segmentation task. So far, this method is still a field worthy of researchers to explore.

**Table 3 micromachines-13-02197-t003:** Summary of the application of deep learning algorithms in breast cancer histopathology for segmentation.

Model	Strategy	Advantages	Publication
ResNet	Propose segmentation of limited data using rough image-level tags with performance comparable to fully labeled datasets	The main feature of ResNet is the residual block, the purpose of the residual block is to preserve the characteristics of the parameters before the current layer is trained and to pass these parameters onto the subsequent layers together with the trained data	[84]
FCN	Propose a fast segmentation method for breast cancer metastases in pathological images	The FCN replaces the fully connected layer behind the traditional CNN with a convolutional layer so that the output of the network will be a heat map rather than a category; at the same time, the image size is recovered using upsampling in order to address the reduction in image size due to convolution and pooling	[85]
	Propose an automatic method for detecting mitosis		[86]
	Describe a method to automatically segment nuclei from hematoxylin and eosin (H&E)-stained histopathology data with fully convolutional networks		[87]
	Use annotated datasets to create accurate models		[60]
	Propose a histopathological tissue analysis framework based on deep learning and verifies its universality and model generalization under different data distributions		[88]
U-Net	Use histopathological images obtained with hematoxylin and eosin staining for biopsy samples for the diagnosis and segmentation of breast cancer	U-Net networks are able to use valid labeled data more effectively from a very small number of training images, relying on data augmentation	[89]
	Address the task of tissue-level segmentation in intermediate resolution of histopathological breast cancer images		[90]
	Propose a deep learning framework consisting of high-resolution encoder paths, pyramidal pooled bottleneck modules in porous space, and decoders		[91]
	Investigate whether it is possible to further improve the performance of the classifier model at the patch level by integrating multiple extracted histological features into the input image		[92]
CNN	Improve the performance of current Simple Linear Iterative Clustering (SLIC) algorithm to achieve hyperpixel segmentation of high-dimensional features		[93]
	Use a pretrained convolutional neural network (CNN) for segmentation and then another Hybrid-CNN for classification of mitoses		[94]
	Identify a useful cell segmentation approach with histopathological images that uses prominent deep learning algorithms and spatial relationships		[95]
	Propose a framework that combines the effectiveness of attention-based encoder–decoder architecture with an empty space pyramid pool with efficient dimensional convolution (kide-Segnet)		[96]
	Propose a deep learning model for automatic segmentation of complex cores in tissue images by encoder-decoder structure		[97,98]
Transformer	Transformer-encoded global features improve U-Net segmentation performance	Transformer model can be used to encode the global features of pathological images and can improve the performance of current algorithms in many fields	[79]

### 3.3. Disease Classification Based on Breast Pathological Images

Classification of medical images by defining the anatomical or pathological features distinguishes certain anatomical structures or tissues. Classification tasks can include many applications in determining the presence of disease, including the identification of tumor types. Deep learning is often used with medical images to classify target lesions into two or more categories. Binary classification refers to distinguishing between breast cancer tissue slices and normal breast tissue slices in the pathological tissue slice dataset used. Multiclass classifications divide pathological tissue slices of the breast into multiple categories using deep learning algorithms based on requirements. Common classifications of breast tissue slices are normal, benign, in situ carcinoma, or invasive carcinoma. In general, the accuracy of the binary classification task is higher than that of the multiclass classification task. Among some existing deep learning classification algorithms, the classification accuracy has reached or even exceeded that of pathologists [43]. Next, we will introduce some typical models that work particularly well for accuracy and performance. Table 4 shows the application of deep learning algorithm in histopathological classification of breast cancer. The application of deep learning algorithm in histopathological classification of breast cancer was classified according to the types of models, and the strategies were summarized.

Convolution neural network is the most widely used method in breast cancer pathological image classification task. [46,99,100,101,102,103,104,105,106,107,108,109] Roy et al. [110] developed a patch-based classifier (PBC) through a convolutional neural network (CNN) for automatic classification of breast cancer histopathological images (Figure 5A). They used two methods: one patch in one decision (OPOD), and all patches in one decision (APOD). Gandomkar et al. [111] proposed a framework MuDeRN (multicategory classification of breast histopathological images using deep residual networks) (Figure 5B). The 152-layer RsetNet is used to sort tasks to improve deeper network optimization for higher model accuracy. Vesal et al. [112] proposed a method based on metastatic learning to divide histological images of breast cancer into four subtypes: normal, benign, carcinoma in situ, and invasive carcinoma(Figure 5C). Migration learning migrates ImageNet’s pretrained parameters into InceptionV3 and RestNet50 and removes the last five layers of the network model to obtain global information. Alom et al. [113] proposed a method for breast cancer classification using the Inception recurrent residual convolutional neural network (IRRCNN) model, which is a combination of the Inception network (Inception-v4), the residual network (ResNet), and a convolutional neural network (RCNN), which provides advantages to DCNN models that exhibit superior performance in object recognition tasks (Figure 5D). Sudharshan et al. [109] proposed a weakly supervised learning framework based on multi-instance learning for classification of breast pathological images. This method does not need to label each instance so it significantly alleviates the problem of difficult labeling of pathological images.

In the field of medical image analysis, transformer-based methods were first used to process disease classification tasks and produced significant results. [114,115,116,117,118,119,120,121,122] There have also been many significant advances in the classification of pathological images of breast cancer. Alotaibi et al. [119] designed an integrated model based on VIT and DeiT to classify pathological images of breast cancer tissues and achieved an accuracy of 98.17 on BreakHis public dataset. However, this method requires pretraining on large-scale datasets and model fine-tuning to alleviate the data hunger of the transformer, which will obviously increase the training cost of the model and limit the scope of use. Shao et al. [120] used the global characteristics of the transformer to build the relationship between instances in order to capture the context information so as to improve the performance of multi-instance learning on the breast whole-slide image. They achieved 93.09% of AUC’s binary classification performance on CAMELYON16 dataset. Chen et al. [121] proposed the Multimodal Co-Attendance Transformer (MCAT) architecture, which aims to build the relationship between WSI and genomic features and use it in survival analysis tasks. This method has proved effective on different cancer datasets, including breast cancer datasets. Chen et al. [122] proposed a multi-scale vision transformer model (GasHis Transformer) for gastric cancer tissue image classification. The author also verified the effectiveness of this method on the breast pathology image dataset. He et al. [123] proposed Deconv-Transformer (DecT), which incorporates the color deconvolution in the form of convolution layers, and uses a self-attention mechanism to match the independent properties of the HED channel information obtained by the color deconvolution. In [124], DCET-Net (based on two backbone streams of CNN and transformer) was proposed, which utilizes CNN stream to focus on the local deep feature extraction of histopathological images, while through the Transformer stream, it enhances the global information representation of images.

The capsule network proposed by Geoffrey Hinton also has some valuable exploration in this field. Anupama et al. [125] used capsule network with preprocessed histology images, which demonstrates that preprocessing data and tuning parameter can improve the performance of conventional architectures. Wang et al. [126] used FE-BkCapsNet based on deep feature fusion and enhanced routing, which combines the advantages of CNN and CapsNet, and the classification performances are better than that of BkNet and CapsNet. However, it is a very time-consuming methodology to classify based on capsule features and convolution features extracted in two parallel channels. Iesmantas et al. [127] developed convolutional capsule network for classification of four types of images of breast tissue biopsy when hematoxylin and eosin staining is applied, but regularization was not taken into consideration.

**Table 4 micromachines-13-02197-t004:** Summary of the application of deep learning algorithms in breast cancer histopathology for disease classification.

Model	Strategy	Advantages	Publication
Deep Belief Network (DBN)	Propose a new patch-based deep learning method PA-DBN-BC for breast cancer detection and classification in histopathological images		[128]
Deep Neural Network (DNN)	Propose a new feature extractor, Deep Manifold Reservation Autoencoder, for automatic classification of breast cancer histopathological images	Deep Belief Network is a probability generation model, which has important value in the early application of deep learning methods	[129]
Generative Adversarial Network (GAN)	Explore whether a deep learning algorithm can learn objective histologic H&E features	GAN can be used as a means of data enhancement to alleviate the problem of insufficient data of pathological images of breast cancer	[103]
Visual Geometry Group Network (VGG)	Discuss and compare the task of automatic amplification in breast cancer detection based on multiple classification	VGG is a deep neural network proposed in 2014, which provides rich deep features for early breast pathological image research	[130]
Recurrent Neural Network (RNN)	Present a deep learning model to classify hematoxylin¨Ceosin-stained breast biopsy images into four classes	RNN can construct the context between pathological image features and be used for prediction of slide-level diagnostic results	[131]
	Propose a second-order multi-instance learning approach that stacks adaptive aggregators by attentional mechanisms and recurrent neural networks (RNN) for histopathological image classification		[132]
Inception	Compare different machine learning methods for classification and evaluation of breast cancer tumors		[133]
	Propose a depth model based on computer-aided transfer learning as a binary classifier for breast cancer detection		[134]
	Propose a method for diagnosing breast cancer as benign or malignant in magnification specific binary (MSB) classification		[135]
Dynamic Convolution Neural Network (DCNN)	Propose an efficient deep convolutional neural network classification model for fast back propagation learning	DCNN can adaptively adjust the convolution kernel parameters according to the input data, enhance the feature expression ability of the model, and specifically solve the tasks related to breast pathological images	[136]
	Develop a deep learning model biopsy microscopic image cancer network (BMIC_Net) for multiple classification of BC		[137]
	Propose two efficient models based on deep transfer learning to improve the binary and multiclassification systems		[138,139]
Convolution Neural Network (CNN)	Propose a new deep architecture based on self-integration to leverage semantic information from annotated images and explore information hidden in unlabeled data		[140]
	Propose an analysis and synthesis model learning method with novel algorithms and search strategies to classify images more effectively		[141,142,143,144,145,146,147,148,149,150]
	Propose a set of training techniques and use image processing techniques to improve the performance of CNN-based models in breast cancer classification		[143,151,152,153,154,155,156,157]
Deep residual network (ResNet)	Present a deep neural network which performs representation learning and cell nuclei recognition in an end-to-end manner		[158]
	Propose an automatic multiclassification method for breast cancer histopathological images based on metastasis learning		[159]
	Present a method that employs a convolutional neural network for detecting tumor on entire-slide images		[59,130,136,160]
	Propose a breast cancer multiclassification method using a proposed deep learning model		[106,113,137,161,162,163,164,165,166,167,168]

Thus, through research and analysis, we found that the classifier proposed by [110] first predicts the class label of each input patch by OPOD technique and then predicts the whole-image label by APOD technique. At the same time, the number of filters and kernel size of each layer are adjusted so that the number of trainable parameters is smaller than the number of samples and overfitting can be prevented. The authors’ proposed framework in [111], MuDeRN, first trains a deep residual network (ResNet) to classify patches in images as benign or malignant. Images classified as malignant were then subdivided into four cancer subgroups, and images classified as benign were divided into four cancer subgroups. MuDeRN classified patients as benign or cancerous with 98.77% accuracy and achieved 96.25% patient-level accuracy across the eight categories. However, for some subtypes with too few cases, MuDeRN’s performance should be investigated on a larger database. In [112], the conventional use of normalized means to deal with color differences, using a different normalized way, showing a good effect but should verify its effect. The author does not use a test set in the training, and the results produced by using only one partition in the training set are hardly convincing. At the same time, the author changed the structure of the end of the network without proving the correctness of the modification by experiment. The method proposed in [113], the IRRCNN model, was used to successfully classify binary and multiple types of breast cancer with constant amplification coefficients. Image and patient-level data were evaluated using different magnifications on publicly available histopathological datasets for breast cancer. Compared with the existing breast cancer classification algorithm, it shows superior performance. Recently, there were also some studies based on attention mechanisms [169] and the use of deep semantic features and image texture features [170] for breast cancer classification. They all achieved good results and provided some references and directions for future research.

From the above results, it can be seen that deep learning has already achieved remarkable results in the field of pathological image classification of breast cancer. How to further improve the performance of deep learning-assisted diagnosis and better provide treatment recommendations based on existing results in pathological image analysis will be the focus of research in the following section. Eventually, we hope to realize a deep learning model that can integrate multimodal data (including medical images, gene sequences, diagnostic reports, drug molecular structure, and other related information) and truly exert the value of deep learning in clinical applications.

### 3.4. Genetic Prediction Based on Deep Learning

WSI is widely used in digital pathology to predict gene mutations, molecular subtypes, and clinical outcomes. Therefore, they are usually divided into patches for training neural networks and prediction models. However, because patch-level tags are usually unavailable, we cannot directly classify each patch. In the past few decades, with the help of the rapid development of high-throughput technologies of microarrays and gene expression analysis technologies, there have been many studies that use gene expression patterns to understand the molecular characteristics of breast cancer. Van de Vijver [171] conducted a preliminary study to effectively predict the prognosis of breast cancer through gene expression profile. They clustered gene expression profile data and correlated them with prognostic values. The integration of gene expression profile data and clinical data may improve the accuracy of prognosis and diagnostic prediction models [172]. In fact, microarray data are high-dimensional, and each patient contains about 25,000 genes. There may be potential relationships between different genes, which may improve the accuracy of prognosis prediction of breast cancer [173]. Many genes related to breast cancer have been identified. Mutation and abnormal amplification of oncogenes and tumor suppressor genes play a key role in the occurrence and development of tumors. For example, two famous breast cancer risk anti-cancer genes, BRCA1 and BRCA2, and human epidermal growth factor receptor 2, also known as c-erbB-2, are important carcinogens in breast cancer, and so on. Table 5 shows the application of deep learning algorithm in genetic prediction of breast cancer.

Khademi et al., proposed the probability graph model (PGM) [172], which predicts and diagnoses breast cancer by integrating two independent microarray models and clinical data. They first applied principal component analysis (PCA) to reduce the dimensions of microarray data and built a depth confidence network to extract the feature representation of the data. At the same time, they also applied structural learning algorithms to clinical data. However, today, inspired by the successful application of deep learning methods in the cv field and the huge contribution of multidimensional data to cancer prognosis prediction, there is a lot of work to directly provide slide-level prediction through deep learning, and digital whole image (WSI) may provide a computationally effective and efficient method to quantitatively characterize the heterogeneity of cancer specimen cell level. Pathologists usually use WSIs to identify nuclear features, diagnose cancer status, and measure histopathological grading of cancer tissues. Preliminary evidence shows that the application of deep learning method can automatically predict the cancer subtypes of various cancers [174], predict the mutations of lung cancer [175] and liver cancer [176], classify mesotheliomas [172], detect DNA methylation patterns [177], estimate the status of human epidermal growth factor receptor in breast cancer [178], and predict the pan cancerous prognosis of patients [179]. However, pan cancer research [180] cannot provide an in-depth description of breast cancer histopathology, mutation, and pathway activity level. At present, DL based on cnn can predict the gene mutation status in H&E-stained WSIs, and it has the potential to improve the prognosis and treatment of cancer by using biomarkers that are currently undetectable to clinicians. Although artificial intelligence cannot completely replace human beings in practice, gene mutation prediction can be used as a prescreening to improve the cost efficiency before next-generation sequencing, thus improving the performance of precision medical treatment.

However, there is still a lack of research related to deep learning that links breast cancer WSI with genes [181]. After our careful search, Wang, Xiaoxiao et al. [182] developed a computer system (DL based on cnn) to predict the molecular marker (gBRCA mutation) of BC through tumor histomorphology analysis. To study whether gBRCA mutation can affect the tumor cell pattern on BCH&E-stained WSIs; Qu, Hui et al. [181] used ResNet followed by a full connection layer with self-attention and maximum pooling to display the weight graph of each tumor tile so as to understand the decision of the classifier and highlight the area that contributes the most to the final prediction. It is proved that the key gene mutation results and biological pathway activities of breast cancer can be predicted through the deep learning classifier of full-slide images; He, Bryan et al. [183] used a complete slide image combined with hematoxylin and eosin (H&E) staining, trained a large number of H&E and IHC-labeled image pair datasets using deep neural networks, and proved the accurate ER receptor state estimation from H&E staining (Figure 6B).

In addition, considering the limitations of the method based on a single information source, such as a lack of nonuniversality, uniqueness, and noise data, multimodal learning is proposed to solve these problems and obtain a final decision by combining relevant information from multiple sources [184,185]. As a kind of multimodal learning, multimodal deep learning [186] proposed a new multimodal deep neural network prognosis prediction for human breast cancer by integrating multidimensional data (MDNNMD). MDNNMD is an effective method to integrate multidimensional data, including gene expression profile, copy number change (CNA) profile, and clinical data with the score level of final prediction results. This method takes into account the heterogeneity between different data types and makes full use of the abstract high-level representation of each data source (Figure 6A).

Petkov et al. [187] accurately predict the prognosis of IDC, which is helpful to determine the individualized adjuvant treatment of breast cancer patients. Lin, Zhiquan et al. [188] proposed and tested WSI preprocessing and feature extraction methods. Combining CAF gene, WSI characteristics, and lymph node status, a multigroup model was established to predict the prognosis of IDC breast cancer patients (Figure 6C).

In the field of digital pathology, unsupervised clustering has been widely used to reduce the dimension of patches to facilitate multi-instance learning (for example, patches from WSI can be immediately installed on the graphics processing unit (GPU)) [189]. This method is also used to derive additional cluster-based characteristics and identify rare events. Dooley et al. [190] and Zhuet al. [191] clustered the plaques and used the frequency of plaques in each cluster as a new feature to predict the rejection of heart transplantation. Similarly, see Abbet et al. [189]. Although various unsupervised clustering applications have been developed in digital pathology, few studies have evaluated the use of unsupervised clustering to identify image patches related to gene mutation. Chen et al. [192] proposed a multi-instance learning method based on unsupervised clustering and developed an in-depth learning model using WSIs of three common cancer types obtained from the Cancer Genome Map (TCGA) to optimize the prediction of genetic mutation.

**Figure 6 micromachines-13-02197-f006:**
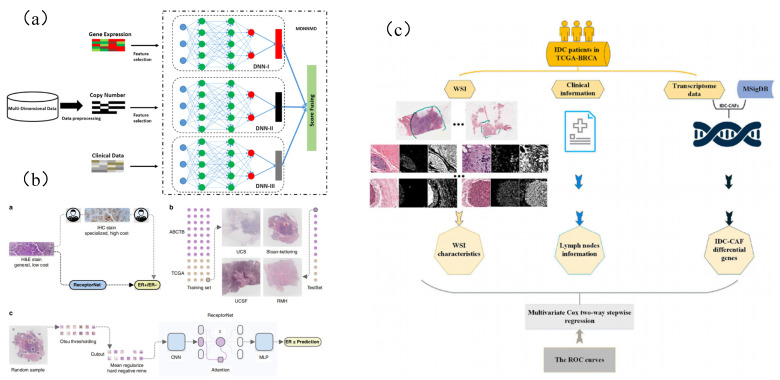
Typical application of deep learning in genetic prediction task. (**a**) MDNNMD model uses multidimensional data to predict the prognosis of breast cancer [186]. (**b**) Estrogen receptor status (ERS) was predicted from the whole-slide image of H&E staining [183]. (**c**) A multi-omics signature to predict the prognosis of invasive ductal carcinoma of the breast [188].

**Table 5 micromachines-13-02197-t005:** Summary of the application of deep learning algorithms in breast cancer histopathology for genetic prediction.

Model	Strategy	Advantages	Publication
Attention mechanism	Weighting different pathological images based on attention mechanism to improve prediction results	Attention mechanism simulates human visual behavior by applying different weights to images. This method can highlight the key areas and non key areas in pathological images, thus improving the prediction results of the model	[181]
	Based on ResNet and attention mechanism, a method for predicting pathological gene subtypes of breast cancer is proposed		[183]
KNN and K-means	Use unsupervised clustering method to reduce the workload of manual labeling by pathologists	Unsupervised	[192]

## 4. Conclusions and Perspective

Deep learning is widely used for the detection, segmentation, and classification of breast cancer pathology and has achieved remarkable results. There are several common networks in the detection, including CNN and RNN, which have reached the level of pathologists in some areas of recognition accuracy. Among these network structures, a CNN-based network structure performs excellently in recognition accuracy. On the Camelyon 16 challenge test set, the CNN-based network NCRF achieved an average FROC score of 0.8096, higher than the previous champion of the challenge. For comparison, the score of professional pathologists is 0.7240. Further comparing the CNN and RNN methods, the RNN can be used to describe the output of the continuous state in time with a memory function, while the CNN is used for static output. Although the RNN can solve problems that the CNN cannot handle, it is not as effective as the CNN. We can integrate the time-series processing method of the RNN into the CNN so that we can combine the output of the continuous state in time to obtain a better result of the image recognition.

There are several common networks in the segmentation, including the FCN, U-Net, RNN, and GAN. Those models have good results in pathological image segmentation, whereas pathological image segmentation methods based on an FCN usually use manually segmented samples at a pixel level as the training dataset and then learn by calculating the loss per pixel. The network structure is affected by subsampling, which makes it difficult to retain meaningful spatial information in the upsampled feature map. In addition to improving the network structure and training learning methods, we can also solve the segmentation problem of pathological cases by defining different loss functions. However, it still relies on the mechanism of comparing the differences per pixel, so its ability to constraint spatial geometric information is very limited. The method of pathological image segmentation based on U-Net is one of the most widely used techniques. The U-Net network can effectively solve the segmentation problem of complex neural structures by capturing global features in the contracting path and achieving an accurate localization in the expanding path. However, the local dependence between pixels is not fully considered, which makes it susceptible to the influence of the external characteristics of the target. We can evaluate the importance of different positional features by combining it with an attention mechanism and assigning weight, and then model the context dependence of the local features. Further comparing the FCN and RNN methods, the network structure and training method adopted by the pathological image segmentation method based on the RNN fully consider the long-term and global dependence between similar pixels, and the ability to capture the spatial and apparent consistency of segmentation marks is enhanced. GAN-based pathological image segmentation methods generally do not need to be modeled in advance; generators and discriminators can choose any structure of the neural network. A GAN model usually leads to poor controllability in the training process and the insufficient stability of the model. Therefore, when using a GAN model to learn the distribution of large-scale pathological data, it is necessary to enhance the stability of the model and its training process. The improvement in the segmentation accuracy of the pathological images of breast cancer will improve the model accuracy of recognition and classification tasks.

In a classification task, the accuracy of multiclassification tasks is usually lower than that of binary classification tasks. Among the deep learning models for classification tasks, the models based on the Inception-V3 series yield a better accuracy in both binary classification and multiclassification tasks. Further comparing the Inception and RestNet methods, Inception requires fewer parameters to set than RestNet. In addition, we can introduce an attention mechanism into a deep learning network to analyze pathological images. The weights corresponding to different scales were learned through the network framework, and then the features of different scales were fused with the attention mechanism to obtain richer features of the pathological images so as to achieve an accurate classification of the pathological images.

The successful application of deep learning in breast cancer pathology has provided pathologists with auxiliary diagnosis methods, which significantly improve the accuracy and efficiency of breast cancer diagnoses. For supervised deep learning algorithms, many labeled training set samples are required to improve the models’ R-squared values. There are few labeled data in the existing public datasets, and the high precision of the pathological sections also makes manual labeling extremely cumbersome. To develop models with higher R-squared values, it is an effective way to obtain more labeled training sets via dataset sharing or using unsupervised deep learning algorithms. In addition, the lack of interpretability of deep learning algorithms has hindered their application in medical diagnosis. It is difficult to understand the features or decision logic of a neural network at the semantic level due to a lack of mathematical tools to diagnose and evaluate the characteristic expression ability of the network (for example, the generalization ability and convergence speed of the depth model). It is also difficult to explain the information processing of different neural network models. When sample data are input into a neural network, we have a hard time explaining the reasons for the predicted results, and optimizing a neural network is difficult. To describe the interpretability of deep learning algorithms, we can start from the internal interpretability based on the model as we can understand the internal operation process of the model while obtaining the output result. We can also start from the interpretability based on the results to infer the operation process of the model from the output results.

With the development of deep learning, AI systems can be built to provide pathologists with auxiliary diagnosis methods. The assisted diagnosis based on deep learning is beneficial to provide more objective and reasonable diagnosis results for patients. Further, AI and healthcare are combined to promote intelligent health management. Intelligent health management is a specific scenario in which artificial intelligence technology is applied to health management for risk identification, virtual nursing, mental health consultation, online consultation, health interventions, and health management based on precision medicine. In the future, pathology image AI needs to further improve its interpretability, such as developing more rational visualization algorithms and adding causal inference to deep learning algorithms.

The medical image analysis method based on deep learning still has many limitations and challenges. At present, the mainstream deep learning method is still the data-driven supervised learning method. Large-scale datasets and fine-grained manual annotation are the key factors for such methods to achieve an excellent performance. This is contrary to the actual clinical environment. The development of deep learning in the field of medical image analysis is limited by the long tail of diseases, the heterogeneity of medical images, and the professionalism of fine-grained labeling. Therefore, how to make full use of large-scale unlabeled datasets, how to mitigate the heterogeneity of multicenter data, and how to make full use of only coarse-grained labels have become the key issues in this field. Unsupervised learning, few-shot learning, image denoising, and other methods will have important value in the future medical image analysis field.

This paper has some limitations. First, this paper focuses on the application of deep learning in breast cancer pathology images, with less description of the innovation of the algorithm and the details of the model. Second, the included articles are mainly based on the prediction of images and lack a focus on multimodal models.

## Figures and Tables

**Figure 1 micromachines-13-02197-f001:**
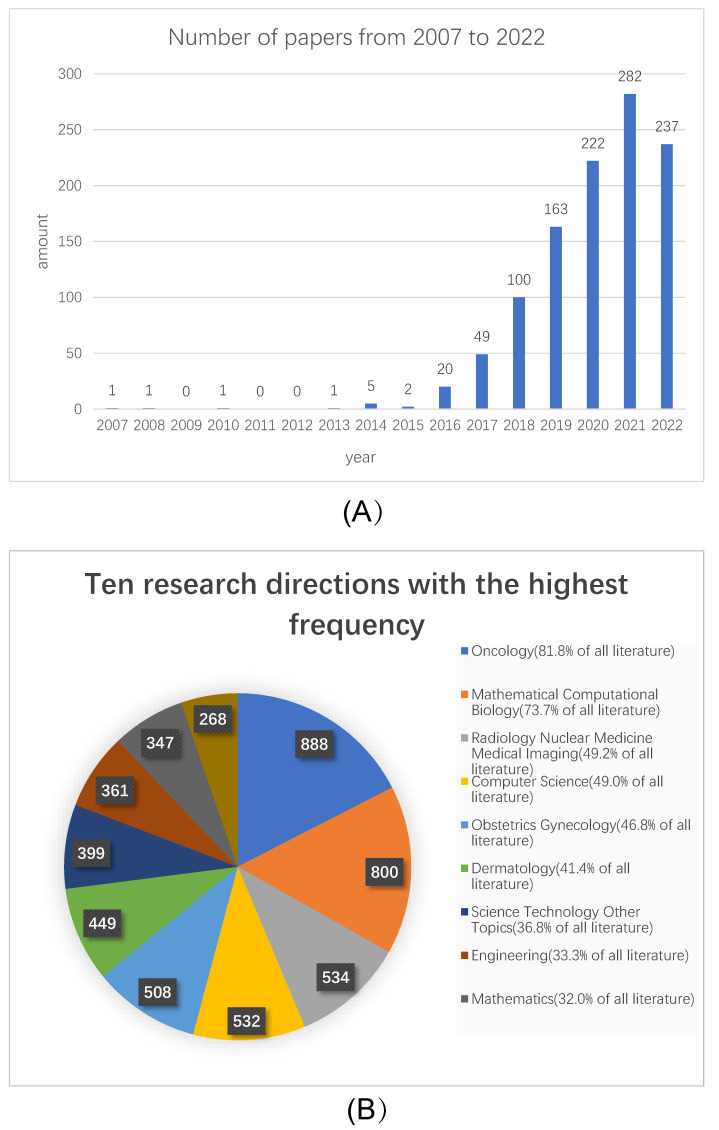
Publication of articles on deep learning of pathological images of breast cancer. (**A**) Year of publication of the article on deep learning in breast cancer pathological images. (**B**) Summary of the themes of review-type articles in articles on the application of deep learning to breast cancer pathological pictures.

**Figure 2 micromachines-13-02197-f002:**
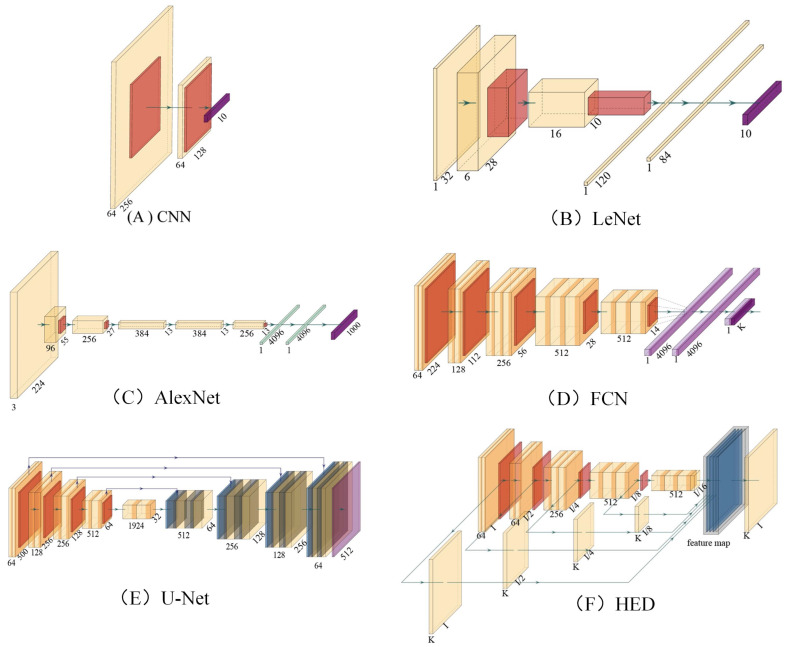
The network structure of the neural network structure commonly used in breast pathological image analysis. These network structure models have been widely applied and performed well in deep learning pathological image classification, segmentation, and recognition tasks. The input size of each layer is shown in the figure. (**A**) Convolutional neural network (CNN). (**B**) LeNet. (**C**) AlexNet. (**D**) Fully convolutional networks for semantic segmentation (FCN). (**E**) UNet. (**F**) Holistically Nested Network (HED).

**Figure 3 micromachines-13-02197-f003:**
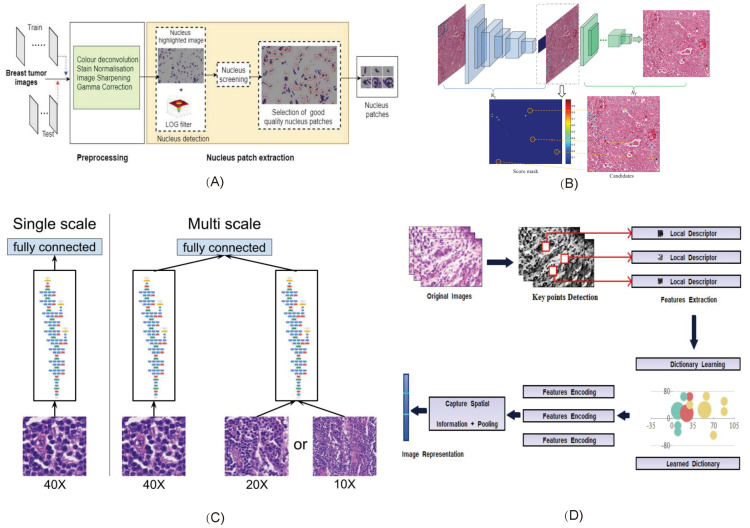
Some typical models of breast lesion detection methods. (**A**) The network structure and model reported by George et al. (**B**) The network reported by Chen et al. The deep cascaded convolutional neural network was used to achieve high accuracy while greatly increasing the speed of analysis. (Reproduced with permission from [Hao Chen], [THIRTIETH AAAI CONFERENCE ON ARTIFICIAL INTELLIGENCE]; published by [PKP Publishing Services Network], [2016].) (**C**) The automatic detection and positioning framework reported by Liu et al., (Reproduced with permission from [Yun Liu], [arXiv]; published by [arXiv], [2017].) (**D**) The network of the automatic classification of breast cancer histological images reported by Bardou et al., (Reproduced with permission from [Dalal Bardou], [IEEE Access]; published by [IEEE], [2018].)

**Figure 4 micromachines-13-02197-f004:**
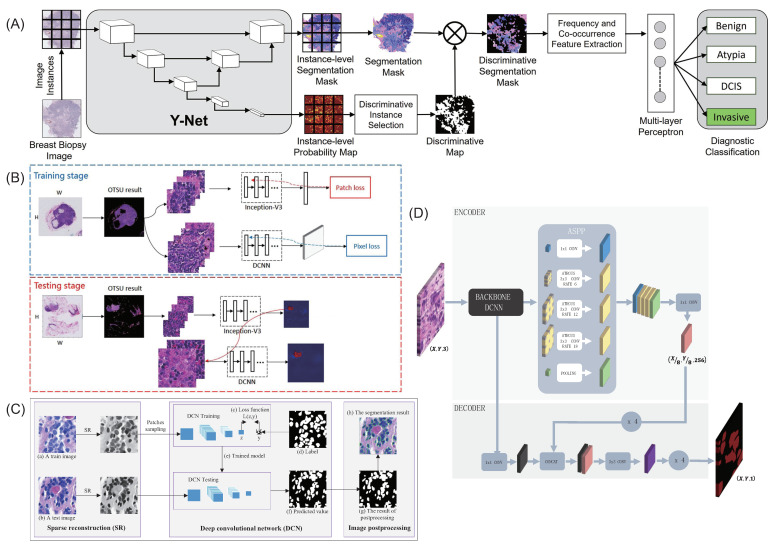
Some typical models of breast pathological image segmentation. (**A**) Overview of methods for detecting breast cancer reported by Mehta et al. (**B**) The fast and refined cancer region segmentation framework v3_DCNN reported by Guo et al. (**C**) The method reported by Pan et al. (**D**) The deep neural network-based pipeline segmentation method reported by Maria Priego-Torres et al.

**Figure 5 micromachines-13-02197-f005:**
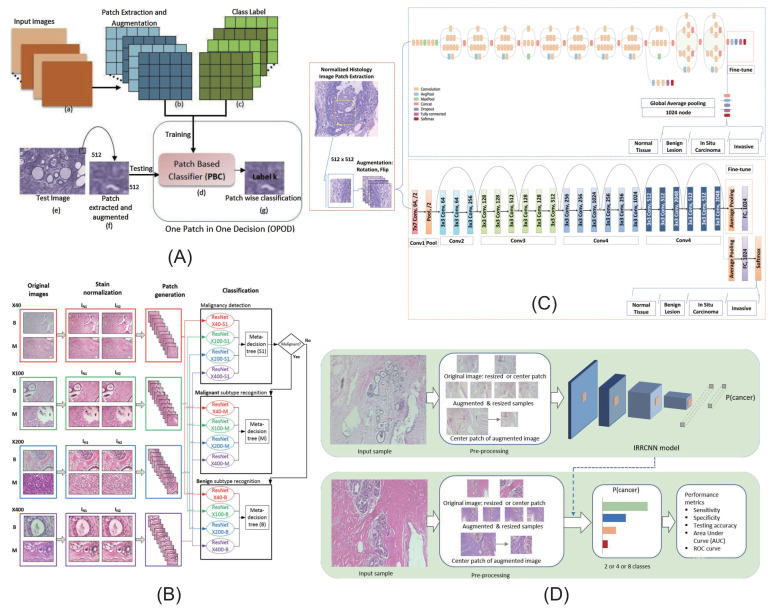
Some typical models of disease classification based on breast pathological images. (**A**) The block diagram of patchwise classification and the block diagram of the CNN architecture reported by Roy et al. (**B**) The steps of MuDeRN reported by Gandomkar et al. (**C**) The workflow of breast histology image classification reported by Vesal et al. (**D**) Alom et al., reported the implementation diagram of the IRRCNN model to identify breast cancer.

**Table 1 micromachines-13-02197-t001:** Common breast cancer pathological image public dataset.

Name	Data Details	
BreakHis	benign	2480
(https://web.inf.ufpr.br/vri/databases/breast-cancer-histopathological-database-breakhis/, accessed on 1 December 2022)	malignant	5429
TCIA	malignant	549
(https://www.cancerimagingarchive.net/, accessed on 1 December 2022)		
GDC Data Portal	malignant	9114
(https://gdc.cancer.gov/access-data/gdc-data-portal, accessed on 1 December 2022)		
Sklearn.datasets	benign	357
(https://scikit-learn.org/stable/modules/generated/sklearn.datasets.load_breast_cancer.html, accessed on 1 December 2022)	malignant	212
BACH (ICIAR 2018 Grand Challenge)	normal	100
(https://iciar2018-challenge.grand-challenge.org/, accessed on 1 December 2022)	benign	100
	malignant	200
Camelyon16	normal	160
(https://camelyon16.grand-challenge.org/, accessed on 1 December 2022)	malignant	240
Camelyon17	normal	160
(https://camelyon17.grand-challenge.org/, accessed on 1 December 2022)	malignant	1240

## Data Availability

Not applicable.

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
