# Peer review of "Application of Deep Learning in Histopathology Images of Breast Cancer: A Review"

_micromachines, 2022, doi:10.3390/mi13122197_

Round 1

Reviewer 1 Report (New Reviewer)

This manuscript reviews state-of-the-art methods and applications of deep-learning analysis in breast cancer pathology. It is thankful that authors cover a wide range of related works with a focus on feature extraction and model development. Notable datasets have been also mentioned to help readers have a better sense about using different datasets.

Detailed comments:

A very important comment. Authors reviewed a large body of feature extraction and segmentation. Yet there is a very active area of developing deep learning pathology to predict key genetic outcomes of cancer. This line of research will advance our understanding of cancer biology and impact treatment decision making in clinics. Unfortunately, authors seem do not mention about this topic. I strongly suggest to at least add one important section to discuss this topic.

To name a few important examples:

https://www.nature.com/articles/s43018-020-0085-8

https://www.nature.com/articles/s41698-021-00225-9

https://www.frontiersin.org/articles/10.3389/fgene.2021.661109/full

In addition, the latest trend of Transformer models has not been discussed in this draft. Transformer models are increasingly important to deep learning as a strong 

State-of-the-art choice for many medical image analysis. Please add more related works in different sections of discussion:

https://arxiv.org/abs/2201.09873

https://link.springer.com/chapter/10.1007/978-3-030-87199-4_6

https://www.sciencedirect.com/science/article/abs/pii/S0031320322003089

Line 43, I disagree that authors’s statement on “medical field has a large amount of high-quality data”. Indeed, collecting and processing large amounts of medical image data for deep-learning analysis is quite challenging. So far, medical community has not even come up with a number comparable to the ImageNet dataset with high-quality annotations. We are in short of support of high-quality, sufficient amounts of training data in medicine. We do have many complex types of data in different departments, across diseases, and organs, but speaking of quantities, they are not sizable enough yet to fully uncover the power of deep learning.

Table 1, can authors add a column to mention the proper use and task for these datasets? This will greatly help readers to quickly consider data use for their own research.

In addition, I suspect the correctness of 33690 as in TCIA. Do you mean the whole slide image samples? Or patient numbers? Please clarify. Also, TCIA and GDC data do have overlap, please add necessary detailed discussions and clarifications on all datasets mentioned in Table 1.

Line 105, citations needed. I doubt which paper (s) do you refer to the first introduction in 2006. Please be cautions on your proper statement and citations. The idea of deep learning actually has long history back to decades ago.

Author Response

Reviewer 2 Report (New Reviewer)

The paper overviews different information fusion techniques used in radiomics The paper is well written. The literature is extensive, however, only 107 are included. Some important papers and deep learning models are missing. Moreover, I cannot find a good discussion on the specific methods and papers in the literature like their disadvantages/limitations. Furthermore, the paper did not mention the criteria for exclusion as well as the searching strategy to conduct this review article. Also, some comments need to be addressed.

.

Introduction:

There are several modalities for breast cancer diagnosis. Please discussed them and the explain why did yo choose histophatology to conduct the survey.

What is the difference between the survey you are conducting and the previous survies that were published.

Why didn’t you use the keyword “histopathology” since your review article title contains this keyword.

Could you please mention the criteria for exclusion from the review? Also, how many papers were intially included and exclude? U searched on google scholar PubMed or what? Your searching strategy?

Why did you select only 107 paper?

Please add a paragraph describing the organization of the paper by the end of the introduction.

Methodology

Please explain in more details the difference of elements (a-f) included in figure 2

Why did not you discuss the methods from [27-57] like results limitations and advantages.

For detection procedure there are important DL models that was not mentioned like YOLO, RCNN,Fast RCNN, Faster RCNN, and SSD please mention then and state the papers that used these models.

In the segmentation section, you did not mention that there are semi-automatic segmentation approaches.

You did not mention any of the deep learning based segmentation methods including  semantic segmentation, Grabcut, U-Net extension, and Graphcut segmentation  etc techniques.

Please add more discussion on the methods on Table 3 [67-77] like their results, limitations abd advantages.

pleas

The authors did not include several important papers for the breast cancer classification and diagnosis from histopatahlogucal images. Could you please include the following:

Automatic Breast Cancer Classification from Histopathological Images." In 2019 International Conference on Advances in the Emerging Computing Technologies (AECT), pp. 1-6. IEEE Computer Society, 2020.

Deep learning for magnification independent breast cancer histopathology image classification,” in 2016 23rd International conference on pattern recognition (ICPR). IEEE, 2016, pp. 2440–2445.

Histo-CADx: duo cascaded fusion stages for breast cancer diagnosis from histopathological images.PeerJ Computer Science,2021, 7:e493.

Breast cancer histopathological image classification using convolutional neural networks,” in 2016 international joint conference on neural networks (IJCNN). IEEE, 2016, pp. 2560–2567.

AUTO-BREAST: a fully automated pipeline for breast cancer diagnosis using AI technology. In Artificial Intelligence in Cancer Diagnosis and Prognosis, Volume 2: Breast and bladder cancer. IOP Publishing

“Deep features for breast cancer histopathological image classification,” in 2017 IEEE International Conference on Systems, Man, and Cybernetics (SMC). IEEE, 2017, pp. 1868–1873.

 “Breast cancer multi-classification from histopathological images with structured deep learning model,” Scientific reports, vol. 7, no. 1, p. 4172, 2017.

 “Histopathological breastimage classification using local and frequency domains by convolutional neural network,” Information, vol. 9, no. 1, p. 19, 2018.

 Classification of breast cancer based on histology images using convolutional neural networks,” IEEE Access, vol. 6, pp. 24 680–24 693, 2018.

 “Multiple instance learning for histopathological breast cancer image classification,” Expert Systems with Applications, vol. 117, pp. 103–111, 2019.

“Automated diagnosis of breast cancer using wavelet based entropy features,” in 2018 Second International Conference on Electronics, Communication and Aerospace Technology (ICECA). IEEE, 2018, pp. 274–279.

Classification of breast cancer histopathology images based on adaptive sparse support vector machine." Journal of Applied Mathematics and Bioinformatics 7, no. 1 (2017): 49.

I can see that the authors did not properly discuss the methods  [82-130] used in the papers included in this section. Please add a good discussion on the papers in this section

please include the limitations and disadvantages of each method.

I believe that the authors should discuss recent deep learning models like capsuleNet and transformers and inlucde these papers:

Breast cancer classification using capsule network with preprocessed histology images. In2019 International conference on communication and signal processing (ICCSP) 2019 Apr 4 (pp. 0143-0147). IEEE.

Automatic classification of breast cancer histopathological images based on deep feature fusion and enhanced routing. Biomedical Signal Processing and Control. 2021 Mar 1;65:102341.

Convolutional capsule network for classification of breast cancer histology images. InInternational Conference Image Analysis and Recognition 2018 Jun 27 (pp. 853-860). Springer, Cham.

 Deconv-transformer (DecT): A histopathological image classification model for breast cancer based on color deconvolution and transformer architecture. Information Sciences. 2022 Aug 1;608:1093-112.

 DCET-Net: Dual-Stream Convolution Expanded Transformer for Breast Cancer Histopathological Image Classification. In2021 IEEE International Conference on Bioinformatics and Biomedicine (BIBM) 2021 Dec 9 (pp. 1235-1240). IEEE.

An Ensemble Model for Breast Cancer Histopathological Images Classification. arXiv preprint arXiv:2211.00749. 2022 Nov 1.

Conclusion

Please add limitations and challenges of deep learning methods.

Also,add the future directions of research

Round 2

Reviewer 1 Report (New Reviewer)

It is appreciated for authors to revise the draft based on comments.

Author Response

Thank you for your comments on improving the paper!

Reviewer 2 Report (New Reviewer)

I would like to thank the authors for addressing most of my comments. However, some minor comments need to be properly addressed.

how many papers were initially included and excluded? 

I asked the authors to  explain in more details the difference of elements (a-f) included in figure 2 and they responded that "We will also further sort out the relevant content of the typical network structure and add it to the paper as soon as possible." Please sort out the relevant content of the typical network structure as Figure 21 now is vague.

Author Response

This manuscript is a resubmission of an earlier submission. The following is a list of the peer review reports and author responses from that submission.

Round 1

Reviewer 1 Report

The authors present a review on machine learning applications on histopathological images in breast cancer. The work is mostly well referenced, but requires significant changes to achieve a publishable standard, and thus should be rejected and resubmitted after being improved.

In particular, many sections are simply paraphrased abstracts of previous works, with little discussion and explanations as to the significance and why the previous work is included in this review. These sections begin typically as “Author et al. proposed (method A)…”. These sections as is, do not represent a significant contribution to the scope of the journal. This is particularly apparent in Section 3.1 on Page 7.

The inline citations of author’s names are also incorrect and inconsistent. They should all be <First Author Surname> et al. Please amend.

Acronyms are not always defined prior to first use, example, “CAD” on Page 2, Line 55. Numerous examples of this also exist in other sections. Please amend.

References are missing from the following statements:

Page 1, Line 22 - … died of breast cancer.

Page 1, Line 34- … diagnostic results.

Page 3, Line 72 - … and complex.

Page 5, Lines 107 to 113.

Page 6, Line 129 - … morphological measurements.

Page 6, Line 132 - … detection accuracy.

In general, claims made are not always backed by references either agreeing or disagreeing with the authors’ statements.

Section 2, Datasets, Pages 3 and 4

There is no link as to why this section is here. This section is summarised in Table 1, and could simply be referred to wherever appropriate.

Figure 1, Page 2

This figure lacks axis labels for both Parts A and B. Part B needs significant revision to be readable and is far too large. The text in Part B is too small and indents are inconsistent, while the coloured boxes are too large. There is also no total for the numbers of publications. The caption needs more detail and only states what the figure is.

Tables, All Pages

All tables need significant revisions. Spacing between text is a significant issue and does not aid readability. Consider using bullet points and reformatting. Acronyms can then be reused especially in Table 4. Also consider keeping a consistent order of algorithms.

On Page 7, Line 151 and Page 9, Line 184 and 185, the sentences used imply that the authors have produced some of the works presented in this review. This is not the case, and the sentences are somewhat misleading and need to be reworded.

Page 17, Line 370

It is unclear why there is a “software” contribution in the author list, as no software has been coded in this review.

Reviewer 2 Report

Authors need to incorporate the following contents. 

1. Search strategy, inclusion and exclusion criteria need to be provided. 

2. Some of the recent deep learning works have not been discussed. Please find few such research works. All such works need to be included.  

Classification of breast cancer from histopathology images using an ensemble of deep multiscale networks

 Breast cancer histopathological images classification based on deep semantic features and gray level co-occurrence matrix

3. All pretrained networks need to be discussed together as one section.

4. Discussion and current research gaps need to be clearly reported. 

Reviewer 3 Report

This paper only contatins a list of articles, without comprehensive and insightful analysis, even with some confusions adout deep learning.

1. For each dataset, the authors should list and analyse those publications, who adopted the dataset, with corresponding methods and (qualitative and quantitive) results.

2. Figure 2, there are mistakes. Pictures and the title are mismatched. Moreover, what’s the defference between AlexNet, LeNet, FCN and CNN? In my opinion, they are all CNN.

3. Detection of breast lesions, Figure 3 (B) (Ref.17) is classification, not detection.

4. The manner of Figure 3 and Table 2 is not good, only copying the figures and descriptions from the original papers, without comprehensive and insightful analysis. Also Figure 4 and Table 3, Figure 5 and Table 4.

Reviewer 4 Report

This is a comprehensive review paper.  Can the authors discuss future directions for the advancement of deep Learning in this field? Also, add some limitations to this review article.